# Ancient segmentally duplicated *LCORL* retrocopies in equids

**Kevin Batcher**[1], **Scarlett Varney**[1], **Terje Raudsepp**[2], **Matthew Jevit**[2], **Peter Dickinson**[3], **Vidhya Jagannathan**[4], **Tosso Leeb**[4], **Danika Bannasch**[1] *

1 Department of Population Health and Reproduction, University of California Davis, Davis, CA, United States of America, 2 Veterinary Integrative Biosciences, School of Veterinary Medicine and Biomedical Sciences, Texas A&M University, College Station, Texas, United States of America, 3 Department of Surgical and Radiological Sciences, University of California Davis, Davis, CA, United States of America, 4 Institute of Genetics, Vetsuisse Faculty, University of Bern, Bern, Switzerland

* dlbannasch@ucdavis.edu

**Data Availability Statement:** All data are available in the main text or the supplementary materials. The retroCNV insertion sites in bigBed format and a track hub for the UCSC Genome Browser are

## Abstract

LINE-1 is an active transposable element encoding proteins capable of inserting host gene retrocopies, resulting in retro-copy number variants (retroCNVs) between individuals. Here, we performed retroCNV discovery using 86 equids and identified 437 retrocopy insertions. Only 5 retroCNVs were shared between horses and other equids, indicating that the majority of retroCNVs inserted after the species diverged. A large number (17–35 copies) of segmentally duplicated Ligand Dependent Nuclear Receptor Corepressor Like (*LCORL*) retrocopies were present in all equids but absent from other extant perissodactyls. The majority of *LCORL* transcripts in horses and donkeys originate from the retrocopies. The initial *LCORL* retrotransposition occurred 18 million years ago (17–19 95% CI), which is coincident with the increase in body size, reduction in digit number, and changes in dentition that characterized equid evolution. Evolutionary conservation of the *LCORL* retrocopy segmental amplification in the Equidae family, high expression levels and the ancient timeline for *LCORL* retrotransposition support a functional role for this structural variant.

## Introduction

Long interspersed nuclear element 1 (LINE-1) is one of the autonomous transposable elements still active in mammalian genomes [1]. LINE-1 encodes two functional proteins: an mRNA binding protein and a combined reverse transcriptase endonuclease. These LINE-1 proteins function to reverse transcribe and insert mRNA copies of LINE-1 into the genome in a process called target-site primed reverse transcription [2, 3]. One hallmark of LINE-1 mediated retro-transposition is the duplication of short segments of genomic DNA at the insertion site, called target site duplications (TSD) [4, 5]. While LINE-1 proteins preferentially act on LINE-1 mRNA, LINE-1 proteins can also create new genomic copies of short interspersed nuclear elements (SINE) as well as copies of other cellular mRNAs, which are referred to as retrocopies or processed pseudogenes [6, 7]. Because retrocopies are derived from processed mRNA, they have a poly(A) tail and lack the introns and regulatory elements present within the parent

available at GitHub: https://github.com/klbatcher/retroCNV_insertions.

**Funding:** This work was funded by Maxine Adler Endowed Chair Funds. The funders had no role in study design, data collection and analysis, decision to publish, or preparation of the manuscript.

**Competing interests:** The authors have declared that no competing interests exist.

gene, aspects which distinguish retrocopies from their parent genes [8]. Most mammalian reference assemblies have thousands of retrocopies which are the products of ancient LINE-1 mediated retrotransposition events and no longer code for functional proteins [9]. Whereas these ancient retrocopies tend to be fixed within species, LINE-1 is still active and capable of inserting new retrocopies [10, 11]. These recently inserted retrocopies vary between individuals within and between species, resulting in what have been referred to as retrocopy number variants (retroCNVs) [12, 13].

While retroCNVs are not typically identified from whole genome sequencing (WGS) data by most common variant calling programs, techniques which take advantage of the differences between the retrocopy insertion and the parent gene of origin can be used to identify retroCNVs, with the "gold standard" of retroCNV discovery requiring identification of the retroCNV parent gene and characterization of the insertion site [2, 14]. Estimates of gene retrotransposition rates have varied due to the difficulty in identifying retroCNVs and the varying number of active LINE-1 between species [11]. However, an analysis of multiple high quality human genome assemblies indicated retroCNVs are more common than previously believed [15]. Additional studies using high coverage WGS data identified 1663 retroCNV parent genes in mice [14] and 503 in human populations [16], while 1911 retroCNVs from 1179 parent genes were recently identified in canids [17]. A recent study in Thoroughbred horses found 62 retroCNV parent genes, although the insertion sites were not identified [18].

Retrocopies are often presumed to be nonfunctional and designated as pseudogenes, yet there is a growing body of evidence indicating that retrocopies are commonly expressed and functional [14, 19, 20]. These expressed retrocopies have been implicated in various diseases including cancer and neurodegenerative disorders in humans [21]. Older retrocopies that no longer code for functional proteins can act as regulatory long non-coding RNAs (lncRNA) which alter the expression or translation of the parent gene [21]. For example, elephants have around 20 transcriptionally active, segmentally duplicated *TP53* retrocopies which, despite being truncated and no longer coding for a fully functional TP53 protein, are thought to play a role in DNA response to damage and cancer resistance [22, 23]. Segmental or tandem duplications of genes and LINE-1 mediated gene retrocopies are both common evolutionary mechanisms across species [24]. While most retrocopies present within reference assemblies are ancestral and have accumulated loss-of-function variants [9], retroCNVs tend to be the result of much younger retrotransposition events and thus typically still code for functional proteins [17]. In dogs, multiple fully functional and expressed *FGF4* retrocopies have been identified based on their association with skeletal dysplasia [25, 26]. Retrocopy insertions may also have functional effects on the expression of nearby genes or form chimeric transcripts when inserted within introns of other genes [12, 13]. While the retroCNVs identified in natural populations of wild mice were found to be under strong negative selection, likely due to deleterious effects [14], significant population differentiation by breed was observed for many of the retroCNVs in dogs, possibly as a consequence of artificial selection by breeders [17].

Horses belong to the order Perissodactyla which also includes the other odd-toed ungulates, such as tapirs and rhinoceroses. Equids (members of the family Equidae) originated ~55 million years ago (MYA) in North America as small forest browsers with four toes [27]. They persisted relatively unchanged until significant climate changes in the mid-Miocene 18–15 MYA, when changes in habitat allowed for the radiation and expansion of species. Morphological characteristics of modern equids developing during this time included larger body size, digit fusion, and hypsodonty (changes to the teeth to allow grinding of grasses) [28]. The genus *Equus* emerged in North America 4.0–4.5 million years ago [29]. Today, all extant equids belong to the single genus, Equus, which is subdivided into three subgenera. *Equus equus*

includes wild and domestic horses, *Equus asinus* includes wild and domestic donkeys, kiangs, and onagers, and *Equus hippotigris* includes all zebras [30].

Because retroCNVs have a propensity to be harmful and artificial selection by breeders may have unintentionally increased the prevalence of these often-deleterious variants, there is a need to characterize the repertoire of retroCNVs in domesticated species. We used a recently developed method which utilizes spliced-mRNA specific sequences and discordant read analysis in order to characterize the landscape of retroCNVs in 86 individual genomes from multiple equid species. While most retroCNVs were not shared between the domestic horses and other equids, many *LCORL* retrocopies were identified in all equids mapping to the same locus, highlighting a segmental duplication containing an *LCORL* retrocopy which is absent from the EquCab3.0 reference assembly. We identified variants specific to the *LCORL* retrocopies and used them to evaluate *LCORL* expression in horse and donkey RNAseq datasets and to estimate the age of the initial retrotransposition event.

## Materials and methods

### RetroCNV parent gene discovery

We adopted a method utilizing mRNA specific 30-mers to identify putative retroCNV parent genes for the EquCab3.0 reference [17]. Briefly, we obtained spliced gene sequences for every gene transcript in the NCBI EquCab3.0 annotation release 103 using Gffread [31]. We then identified mRNA specific 30-mers which are absent from the EquCab3.0 assembly for each gene sequence using Jellyfish [32]. To prevent false positives due to sequencing errors, all 30-mers with an edit distance of 2 substitutions from the EquCab3.0 reference assembly were filtered using mrsFAST [33]. In total, 26,849 genes were used for retroCNV discovery using 5,672,946 unique 30-mers (median 30-mers per gene: 131). We then performed retroCNV discovery using high coverage Illumina paired end whole genome sequencing (WGS) fastq files downloaded from the Sequence Read Archive or European Bioinformatics Institute [34]. The dataset included 75 breed horses, 3 Przewalski's horses and 8 wild Equids (S1 Table). We queried fastq files for the presence of the mRNA specific 30-mers using Jellyfish, and we considered any genes which had greater than 5 mRNA specific 30-mers and at least 10% of the total 30-mers for that gene as putative retroCNV parent genes.

### RetroCNV insertion site identification

We resolved the retroCNV insertion sites using a previously developed pipeline [17]. Briefly, we aligned WGS fastq files to the EquCab3.0 reference assembly [35] using Minimap2 v2.24 with the preset '-ax sr' for Illumina paired end reads [36]. Aligned data were sorted and duplicate reads were removed using samtools [37]. We then used TEBreak to obtain discordant read clusters at putative retroCNV parent gene loci [38]. We visually confirmed all retroCNV insertion sites using IGV [39]. The TEBreak 5' and 3' junction sequences for the retroCNVs are available in S2 Table. To validate a set of retroCNV insertion sites and TSD, we designed three primer genotyping PCR assays as previously described [26] (S3 Table). For genotyping, we randomly selected thoroughbred horses from a DNA repository maintained by the Bannasch lab. These DNA samples were collected. We performed sanger sequencing on an Applied Biosystems 3500 Genetic Analyzer using a Big Dye Terminator Sequencing Kit (Life Technologies, Burlington, ON, Canada). We also analyzed the horse Y chromosome assembly eMSYv3.1 (GenBank accession MH341179) [40] for evidence of retroCNVs that had been predicted to be on the Y chromosome based on sex bias. The retroCNV parent gene sequence was used to query the Y chromosome for the retrocopy using BLAST [41].

### Reference assembly retrocopy analysis

In order to identify retroCNVs within the EquCab3.0 reference assembly, we analyzed the horse retrocopies present in RetrogeneDB [9]. We first converted the retrocopy locations EquCab3.0 using the NCBI remap tool, and then analyzed the WGS data for deletions at these locations using Delly [42]. Deletions which confirmed the retrocopy as a retroCNV were manually confirmed visually using IGV. We also visually analyzed all of the recent (>95% identity, N = 88) retrocopies from RetrogeneDB in the 8 wild equids using IGV to manually determine if they were retroCNVs.

### RetroCNV specific variant analysis

We identified single nucleotide variants (SNV) at the retroCNV parent gene loci from the WGS dataset using bcftools mpileup [37]. For each gene, SNVs which were present only in individuals who were positive for a retroCNV of that gene were attributed to the retroCNV, while SNV which were present in individuals who lacked the retroCNV were considered variants within the parental gene sequence itself. RetroCNVs which were unique to wild equids were excluded from this analysis.

### *LCORL* retrocopy copy number analysis

In order to estimate the total copy number of the *LCORL* retrocopies in each sample, we first calculated the average coverage over a portion of the segmental duplication (Equcab3 Chr9:30186485–30216186) using the samtools depth [37]. This was then divided by the average genome-wide coverage as estimated using samtools idxstats [37].

### *LCORL* expression analysis

We obtained a list of *LCORL* retrocopy specific variants from WGS data using bcftools mpileup at the *LCORL* parent gene locus [37]. We analyzed the *LCORL* retrocopy specific variants in IGV, and used only variants which appeared fixed across all retrocopies to create a masked *LCORL* transcript (S2 File). We downloaded Illumina paired-end RNA-seq data from the Functional Annotation of Animal Genomes (FAANG) archive [43] and aligned them to the masked *LCORL* sequence using HISAT2 [44]. We then used SNPsplit to perform allele specific analysis of *LCORL* in order to differentiate the parental gene transcripts from the retroCNV transcripts [45]. We also aligned RNA-seq data to the full EquCab3.0 reference assembly using HISAT2 and calculated TPM for *LCORL* and *MRPL15* using Kallisto [46]. TPM were then averaged across all sample of the same tissue type. We also performed the same analysis using Illumina paired-end RNA-seq data from a Dezhou Donkey (PRJNA431818) [47] using a separate masked *LCORL* sequence file created with the donkey *LCORL* retrocopy specific variants (S3 File). We obtained normalized expression data in humans for *MRPL15* and *LCORL* from the Human Protein Atlas website (proteinatlas.org) [48]. We also analyzed expression of the *LCORL* retrocopies in long-read RNAseq data from the FAANG initiative [49]. Circular consensus sequence files were downloaded and aligned to EquCab3.0 using minimap2 v.2.24 using the preset '-ax splice:hq' for spliced Iso-seq data, and the aligned files were then visually analyzed in IGV [39].

### Fluorescence *in situ* hybridization

Metaphase and interphase chromosome preparations of horses, donkeys, Przewalski's horses, an African wild ass, a Hartmann's Mountain zebra, a Common zebra, and a Southern white rhino were available from the collection of the Molecular Cytogenetics laboratory at Texas

A&M University. Fluorescence *in situ* hybridization (FISH) was conducted following standard procedures [50] using as a probe a BAC clone 164B3 from horse genomic BAC library CHORI241 (https://bacpacresources.org/) known to contain *LCORL* retrocopies [51]. BAC DNA was labeled with digoxigenin-11-dUTP, using DIG-Nick Translation Mix (Roche Life Science) and hybridized to equid and perissodactyl chromosome preparations. The hybridization sites were detected with DyLight® 594 anti-digoxigenin conjugate (Vector Laboratories, Burlingame, CA, USA) and imaged with a Zeiss Axio Imager M2p fluorescence microscope equipped with a high-resolution progressive scan CCD camera CoolCube 1 and Isis V5.3.18 software (MetaSystems GmbH, Altlußheim, Germany). Chromosomes were counterstained with 4'-6-diamidino-2-phenylindole (DAPI) and identified following cytogenetic nomenclatures for the horse [52] and the donkey [53], and available karyotype arrangements for other species [54]. The same procedure was followed to conduct FISH with the CHORI241 BAC clone 79K4 containing the single copy gene *PLCG2* on horse chromosome 3p12 for comparison to the segmentally duplicated *LCORL* retrocopies.

### Phylogenetic analysis of the *LCORL* retrocopies

We queried two recent, chromosome level wild equid genome assemblies for evidence of the *LCORL* retrocopy via BLAST using the *LCORL* cDNA sequence: the Dezhou donkey (*Equus asinus*) assembly ASM1607732v2 [47], and the plains zebra (*Equus quagga)* assembly UCLA_-HA_Equagga_1.0 [55]. The *LCORL* retrocopy sequences were extracted from the assemblies, aligned to the respective parent gene sequence using MUSCLE [56], and the number of SNPs between the retrocopies and parent gene were counted using SNP-sites [57]. We then aged the *LCORL* retrotransposition using an estimated mutation rate in horses of $7.24 \times 10^{-9}$ mutations/site/generation, with a generation time of 8 years [29, 58]. The Southern white rhinoceros (*Ceratotherium simum*) and South American tapir (*Tapirus terrestris*) genome assemblies [59] were also searched for evidence of the *LCORL* retrocopies with BLAST using the *LCORL* cDNA sequence. To further confirm that no *LCORL* retrocopies were present in other Perissodactylas, we also queried WGS data from two species of rhinoceros (DRR308100, SRR20637451) and two species of tapir (SRR11097180, SRR13167957) via BLAST using a spliced *LCORL* sequence between exon 5 and exon 6 (S4 File). We performed phylogenetic analysis of the *LCORL* parent genes and retrocopies using the previously identified retrocopy specific variants (S6 Table) and the R package pvclust with method.dist set to Euclidian [60]. Information on weight, age, and digit status for the Equidae were obtained from previous publications [29, 61–64].

### Ethics statement

DNA used in this study were extracted from whole blood collected from horses seen at the UC Davis veterinary medical teaching hospital following an Animal Care and Use Protocol (#15963) that was approved by the University of California Davis Institutional Animal Care and Use Committee.

## Results

### RetroCNV discovery in equids

We performed retroCNV discovery using Illumina paired-end WGS data from 86 equids (75 domestic horses, 3 Przewalski's horses and 8 wild equids) aligned to the EquCab3.0 domesticated horse reference assembly (S1 Table). The median genomic coverage across all samples was 21.2x. An initial analysis using spliced-mRNA specific 30-mers identified 693 putative

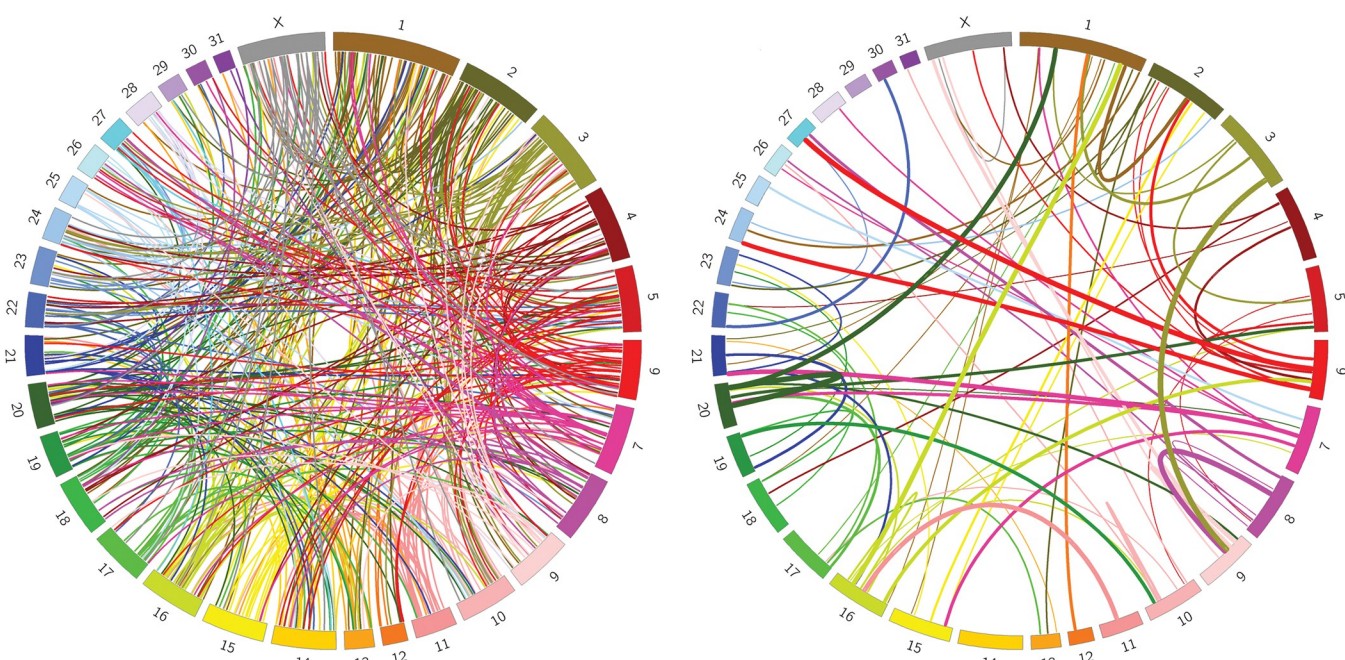

**Fig 1. Circos plots of equine retroCNVs.** Links are colored based on the chromosome of parent gene. Left: All retroCNVs identified in 86 equids (N = 437). Chromosomal locations are based on EquCab3.0. Right: All retroCNVs in 22 Thoroughbreds (N = 81). Thickness of the band indicates how common the retroCNV was in Thoroughbreds.

retroCNV parent genes across all samples. We then identified the retroCNV insertion sites using discordant read analysis focused on these putative retroCNV parent genes. In total, 437 retroCNV insertions from 353 parent genes were resolved (Fig 1, S4 Table). Most horse breeds were represented by only a few individuals, but we used 22 Thoroughbred horses. Many retroCNVs showed high allele frequency within this breed (Fig 1). Many of the retroCNVs (146/437; 33.4%) were inserted within introns of other protein coding genes (S4 Table). We identified the TSD sequence for 316 of the 437 retroCNVs based on overlapping discordant reads. The TSD sequences ranged from 10 to 31 bp with a median length of 17. A genotyping matrix showing which individuals carry which retroCNV is available in S4 Table. Primers were designed for eight retroCNVs, and a set of Thoroughbreds were tested to validate the insertion sites. Sanger sequencing of the PCR products confirmed the presence of retrocopy insertions, and the TSD sequences matched those that were predicted from the WGS data (S5 Table).

The number of non-reference retroCNVs in each individual domesticated horse ranged from 8 to 40 with an average of 22.5 retroCNVs (Table 1; 95% CI 21.3–23.7). The 8 wild equids

**Table 1. Summary of non-reference retroCNVs in equids.**

| Population | Total retroCNV | Average retroCNV | Exclusive retroCNV |
|---|---|---|---|
| All Domesticated Horses (N = 78) | 219 | 22.5 | 191 |
| Thoroughbred Horses (N = 22) | 81 | 22.1 | 9 |
| Przewalski's Horses (N = 3) | 50 | 29 | 23 |
| Wild equids (N = 8) | 223 | 61.1 | 218 |

Average retroCNV is the average number of retroCNVs carried by an individual animal with respect to the genome reference assembly. Differences in sample numbers between these populations will affect these values.

averaged 61.1 retroCNVs (95% CI 55.0–67.2). Nearly half of the retroCNVs (218/437, 49.9%) were exclusive to the 8 wild equids. In the domesticated Thoroughbred horse breed, where we sampled 22 individuals, 5 private retroCNVs were identified. We also analyzed the retroCNVs that were specific to domesticated horses for unique variants which differentiate them from their parental genes. Most (163/219, 74.4%) of the retroCNVs in the domesticated horses did not contain any unique variants, and only one high impact variant was detected, indicating that most retroCNVs still likely code for functional proteins. The list of retroCNV specific variants and their predicted effects is available in S6 Table.

While discordant read analysis failed to identify an insertion site for 340 of the 693 putative retroCNV parent genes, we observed exon-exon discordant reads for only 51 of those 340 putative parent genes, indicating that retroCNVs with unresolved insertion sites were present for those parent genes (S7 Table). Four of these putative parent genes (*ATP6V0C*, *EIF3CL*, *LOC100066257*, *TIRAP*) had discordant reads present only in the male individuals, possibly indicating that the retroCNVs were on the Y chromosome, which is absent from the Equ-Cab3.0 reference assembly. To test this, we analyzed an equine Y chromosome assembly (eMSYv3.1) for the presence of those four retroCNVs, and identified three of them (*ATP6V0C*, *EIF3CL*, and *LOC100066257*).

Because the individual animal used to create the reference assembly (EquCab3.0) may also contain unique retrocopy insertions that are not present in other equids, we analyzed the retrocopies present in the EquCab3.0 assembly itself to determine if there were retroCNVs. In this analysis, we considered any of the EquCab3.0 reference retrocopies that were absent in any samples to be retroCNVs. We identified a total of 32 reference retroCNVs (S8 Table). These 32 reference retroCNVs were absent from all the wild equids, making them horse-specific retroCNVs; additionally, 21 of them were present in every domesticated horse sample. The reference retroCNVs shared 98.3% sequence identity with their parent gene sequences on average and tended to be the full length of the parental protein coding sequence (97.4% on average).

## RetroCNV parent gene analysis

Many of the retroCNV parent genes were genes which are commonly retrotranscribed in other species, including multiple genes encoding ribosomal proteins and *GAPDH*. The 353 retroCNV parent genes were highly enriched for biological processes of translation and peptide synthesis, and molecular functions related to the ribosome and RNA binding (Table 2).

### *LCORL* retrocopy in equids

Only 5 retroCNVs were shared between wild equids and domesticated horses (*MALSU1L1*, *LCORLL1*, *L2HGDHL1*, *SKA3L1*, *PTP4A1L5*). Among them, an *LCORL* retrocopy was

**Table 2. GO analysis of equine retroCNV parent genes.**

| GO biological process | Number | Expected | Fold Enrichment | P value | FDR |
|---|---|---|---|---|---|
| translation | 33 | 5.70 | 5.79 | 3.49E-15 | 5.01E-11 |
| peptide biosynthetic process | 33 | 5.96 | 5.53 | 1.14E-14 | 8.17E-11 |
| peptide metabolic process | 34 | 7.42 | 4.59 | 6.79E-13 | 3.25E-09 |
| cellular macromolecule biosynthetic process | 41 | 11.15 | 3.68 | 2.21E-12 | 7.93E-09 |
| GO molecular function | Number | Expected | Fold Enrichment | P value | FDR |
| structural constituent of ribosome | 22 | 3.32 | 6.63 | 1.44E-11 | 6.41E-08 |
| RNA binding | 46 | 15.56 | 2.96 | 1.12E-10 | 2.49E-07 |
| structural molecule activity | 29 | 9.31 | 3.11 | 1.41E-07 | 6.95E-05 |
| mRNA binding | 19 | 4.39 | 4.33 | 2.23E-07 | 9.01E-05 |

identified which was present in all 86 domesticated horses and wild equids, indicating that it was common to all equids and absent from the EquCab3.0 assembly. Because the *LCORL* gene locus has been associated with height in horses [65, 66] as well as other mammals [67–71], we examined the *LCORL* retrocopies for evidence of function. Analysis of an earlier version of the domesticated horse genome assembly EquCab2.0 indicated that an *LCORL* retrocopy was present on an unplaced contig which is absent from the EquCab3.0 assembly. Discordant reads identified a single insertion site for all samples at EquCab3.0 chr9:30194359–30194380, within an intron of *MRPL15*. However, an abnormally large number of discordant reads were observed for the *LCORL* retrocopy; each equid had on average 750 discordant reads (95% CI 654–847) mapping to the *LCORL* parent gene on chromosome 3 and the same insertion site on chromosome 9, which indicated there were many *LCORL* retrocopies at this location (Fig 2, top; S1 Fig). A previous study had also indicated the presence of multiple *LCORL* retrocopies on chromosome 9 using fluorescence *in situ* hybridization (FISH) [51]. Visual analysis of the short read alignments at the *LCORL* retrocopy insertion site revealed increased coverage in a ~30 kb interval flanking the insertion, indicating that the retrocopy was part of a larger segmental duplication (S1 Fig).

An analysis based on increased coverage per individual indicated that there were between 17 and 35 copies of the segmental duplication carrying the *LCORL* retrocopy in horses (S9 Table). The number of *LCORL* retrocopies appears to vary between individuals and species within the Equus genus; members of the subgenus *Asinus* had the least number of copies on average at 13.7 (95% CI 10.3–16.6), while domestic horses had an average of 25.1 (95% CI 24.4–25.9) and Przewalski's horses were estimated to have 33.2 copies of *LCORL* on average (95% CI 30.0–36.4). We then analyzed additional equine genome assemblies for the presence of *LCORL* retrocopies. The UCLA_HA_Equagga_1.0 zebra genome assembly contains six *LCORL* retrocopy fragments on unplaced scaffolds, indicating that the region had not been correctly assembled. While the ASM1607732v2 donkey genome assembly contains 10 full length *LCORL* retrocopies (S10 Table), only one of the *LCORL* retrocopies was on donkey chr12, which is homologous to horse chr9, and the other nine *LCORL* retrocopies were assembled on donkey chr3. A separate Donkey genome assembly, ASM130575v2, also contained two full length *LCORL* retrocopies on unplaced scaffolds.

In addition to sequence analysis, we visualized the locations and approximate copy numbers of *LCORL* retrocopies in horses and other equids by FISH and confirmed the presence of multiple retrocopies on horse chr9 and its homologs in other equids (Fig 2, bottom). FISH results also indicated individual and species variation in the number of *LCORL* retrocopies and confirmed that the asses have slightly less copies than horses, and zebras (S2, S3 Figs). FISH verified that the ASM1607732v2 donkey assembly had inaccurate placement of the *LCORL* retrocopies on donkey chr3, as all the FISH signals were on donkey chr12, homologous to horse chr9.

The *LCORL* gene encodes multiple different isoforms through alternative splicing. The retrocopy is derived from a transcript which lacks exon 3 and which includes the first of two possible terminal exons (Fig 2, top). This specific transcript matches a predicted transcript variant in mice "*Lcorl* transcript variant x12", where a second start codon in exon 2 is predicted to produce a 512 amino acid product (NCBI: XM_030254341.2). The assembled retrocopy sequences present in the EquCab2.0 and ASM1607732v2 assemblies were all predicted to result in a truncated protein product 8 amino acids in length due to a nonsense variant in exon 4 present in all retrocopies (S1 File). Based on visual inspection of the *LCORL* retrocopy across the 86 sequenced genomes, it had many shared SNVs indicating that it was likely due to an older retrogene insertion (Fig 3A). We performed variant calling across the *LCORL* parent gene locus to identify variants specific to the retrocopies. We detected numerous variants that were

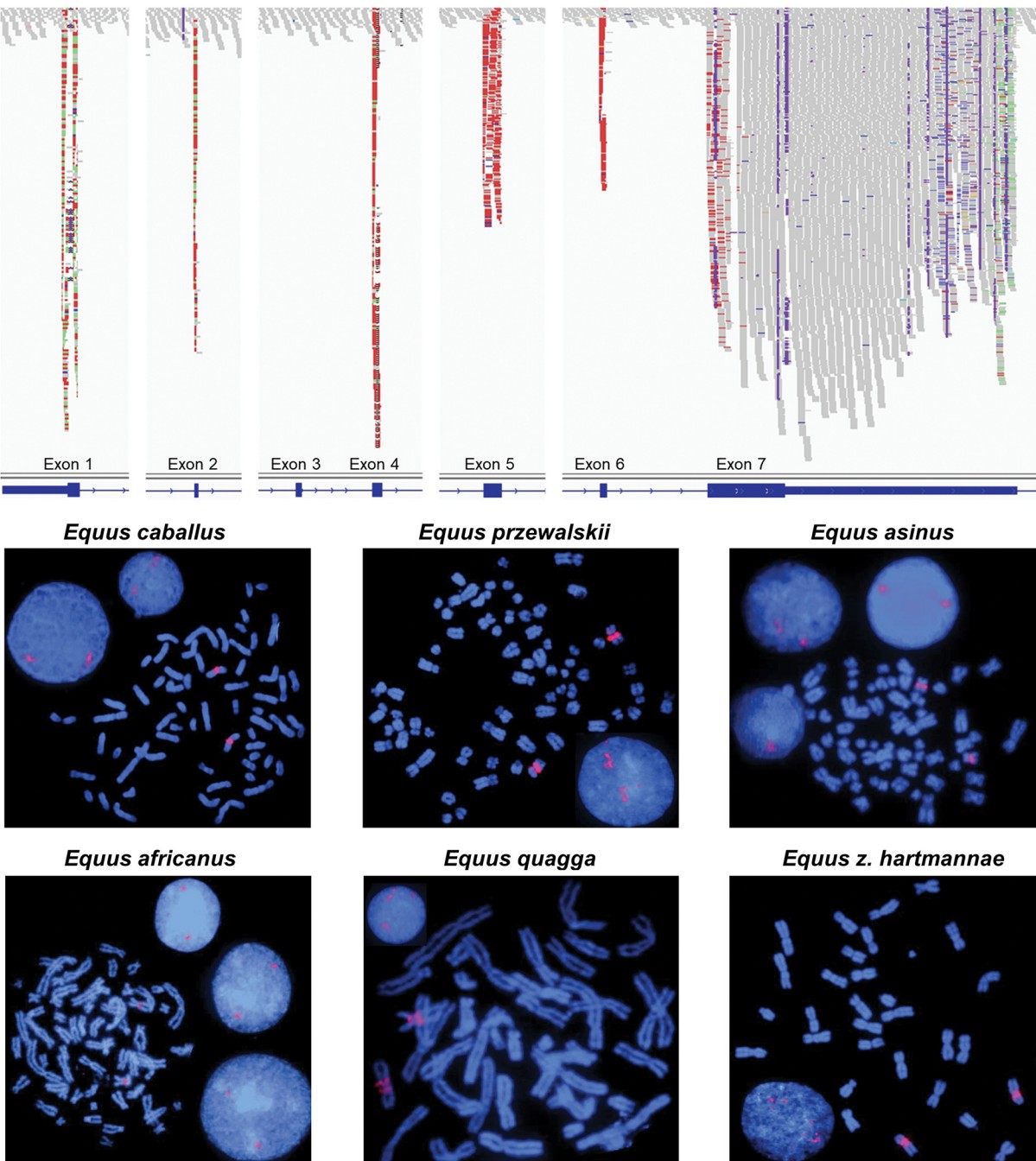

**Fig 2. Large number of expressed *LCORL* retrocopies in equids. Top**: Increased coverage over the exons of the *LCORL* gene in WGS data indicate the presence of several *LCORL* retrocopies in equids. Paired end reads from the *LCORL* retrocopies, highlighted in red, align only to the exons of the parent gene due to the absence of introns in the retrocopies. Reads highlighted in green at the 5' and 3' end of the gene indicate the paired read is aligned on a different chromosome, in this case chr9 where the *LCORL* retrocopy is inserted. Notably, the third exon is missing from the retrocopy transcript. **Bottom**: FISH with horse BAC clone 164B3 in different equids confirm the presence of *LCORL* retrocopies. The red signals in metaphase and interphase chromosomes show the location and approximate copy number of *LCORL* retrocopies.

unique to the *LCORL* retrocopy sequence and which differentiate it from the parent gene, including multiple frameshift mutations, a 33 bp deletion in exon 1 corresponding to the loss of 11 alanine codons, and a nonsense variant in exon 4 (S11 Table). The *LCORL* retrocopy specific variants also differed by species; while 15 variants appeared fixed in all *LCORL* retrocopies in all Equid species, 24 variants appeared unique to and fixed within only the horse *LCORL* retrocopies, and 6 variants appeared in all wild equid *LCORL* retrocopies but were absent from horses. The horse retrocopies also contain a 76 bp deletion in exon 7 (chr3:107,550,405–107,550,481) and a SINE insertion in the 3' UTR (chr3:107,553,402) which are not present in any of the wild equid retrocopies.

## *LCORL* expression analysis

We then analyzed RNAseq data from the Functional Annotation of Equine Genome (FAANG) dataset [43] using the *LCORL* retrocopy specific variants to differentiate the parental gene transcripts from the retrocopy transcripts. The majority of *LCORL* transcripts were derived from the retrocopies (Fig 3B; S4 Fig). This same trend was observed in a donkey RNA-seq dataset, where the majority of *LCORL* expression could be attributed to the retrocopy

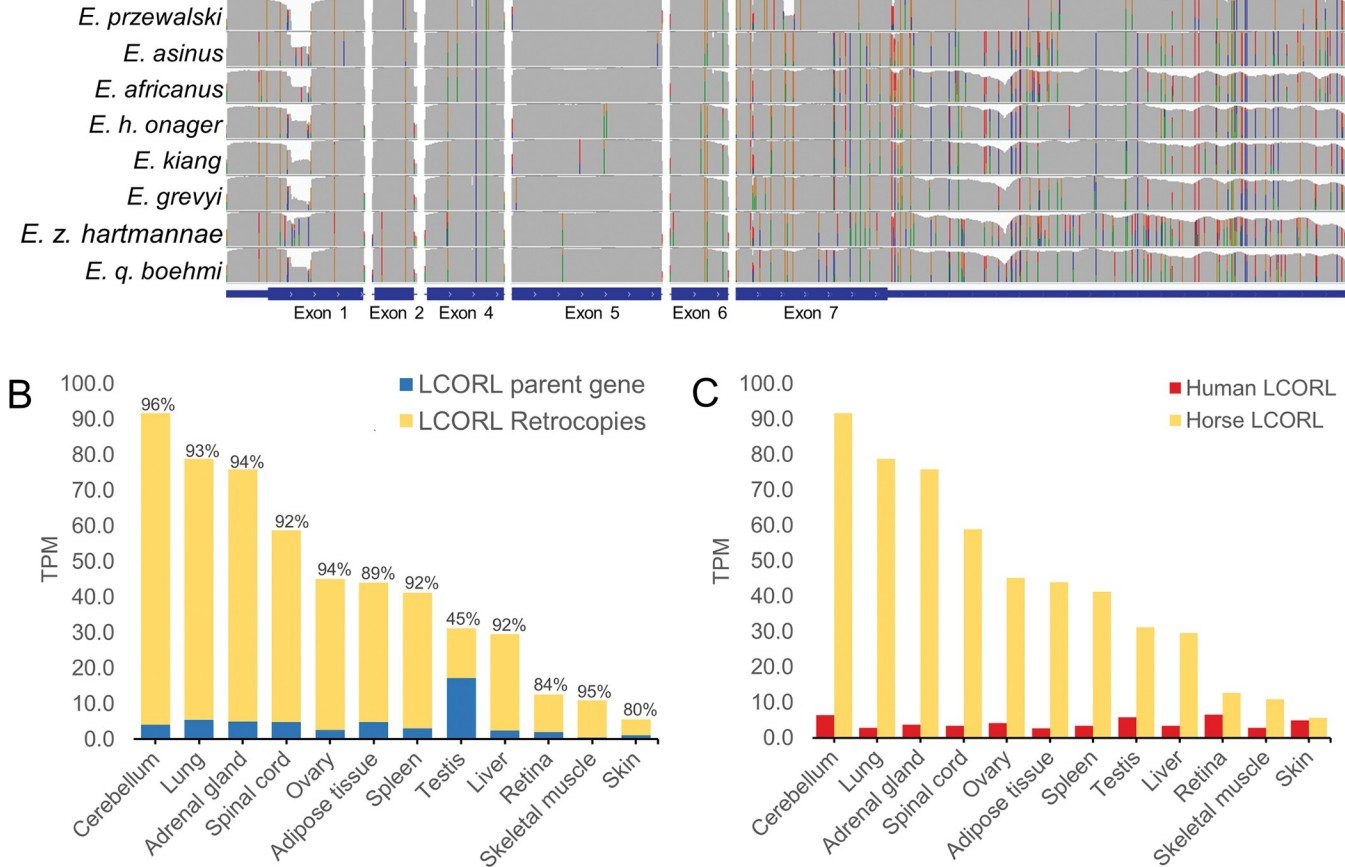

**Fig 3. Functional analysis of the *LCORL* retrocopies.** (**A**) Visual representation of the variation across the *LCORL* gene. The colored lines indicate the presence of SNVs, highlighting the sequence differentiation within the *LCORL* retrocopies between species. Each colored line represents a single nucleotide variant within the retrocopy sequence. Not all variants appear fixed across all retrocopies. (**B**) *LCORL* mRNA expression analysis in transcripts per million (TPM) in horses based on the SNVs identified between the retrocopy and the parent gene indicated that the *LCORL* retrocopies comprised the majority of overall *LCORL* transcripts in all tissue types except testis. (**C**) Overall *LCORL* expression was also increased in horses relative to humans.

(S5 Fig). Overall expression of *LCORL* was also increased in all horse tissues relative to humans (Fig 3C), with an average fold increase in transcripts per million (TPM) across 12 tissues of 11.8 (95% CI 7.4–16.3). The increase in overall expression of *LCORL* in horses was entirely due to expression of the retrocopies (S6 Fig). *MRPL15*, which is also included within the segmentally duplicated region in equids, was also overexpressed in horses compared to humans (S7 Fig; average fold-increase in expression 12.1, 95% CI 8.1–16.1). Tissue specific trends in expression of *MRPL15* and *LCORL* in donkeys and horses are available in S8 Fig. We also analyzed long read RNAseq data to characterize the full length *LCORL* retrocopy transcript. The *LCORL* retrocopy often appears to form a chimeric transcript with nearby *MRPL15*, and antisense transcripts of the *LCORL* retrocopy were observed at the insertion site (S9 Fig).

## Phylogenetic analysis of the *LCORL* retrocopy

Phylogenetic analysis using the *LCORL* retrocopy variants confirms known relationships between the species and supports the hypothesis that the *LCORL* retrocopy insertion predates *Equus* speciation (Fig 4A). Interestingly, rearrangements of the segmental duplication containing the *LCORL* retrocopy and *MRPL15* gene appear to have resulted in gene conversions: we identified several variants in both the parent gene and retrocopy sequence in one species that were absent from other species' parent or retrocopy sequences. No evidence of *LCORL* retrocopies was found in genome assemblies of two other extant members of the order Perissodactyla, the South American tapir (*Tapirus terrestris*) and the white rhinoceros (*Ceratotherium simum*). Additionally, no *LCORL* retrocopies were present within the WGS data of two tapirs (*Tapirus terrestris* and *Tapirus indicus*) and two rhinoceroses (*Rhinoceros unicornis* and *Cerototherium simum*), indicating that the *LCORL* retrocopies were specific to the Equidae family (Fig 4B). This is in line with FISH results showing no *LCORL* hybridization signals in white rhinoceros interphase or metaphase chromosomes (S10 Fig).

Since variants exist within the duplicated copies of the *LCORL* retrocopy at varying allele frequencies, we compared the single haplotype retrocopy sequences present within the EquCab2.0, ASM1607732v2 and UCLA_HA_Equagga_1.0 references assemblies to their parent gene sequences in order to estimate how old the *LCORL* retrocopy was (S12 Table). Estimates

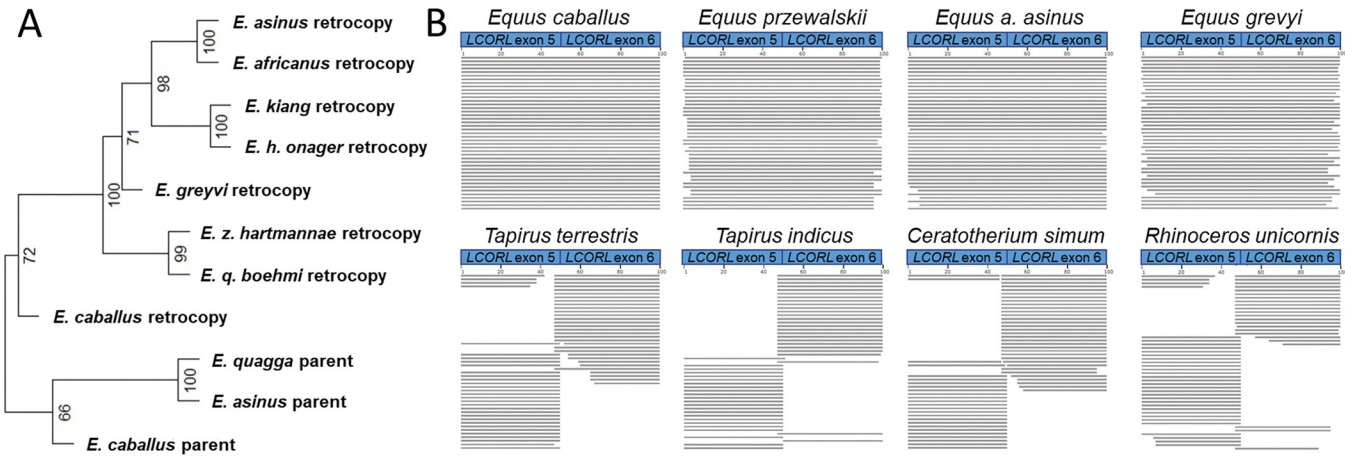

**Fig 4. Across species analysis of *LCORL* retrocopies.** (**A**) SNVs within the *LCORL* retrocopies as compared to the parent *LCORL* gene were used to reconstruct the equine phylogenetic tree indicating that the retrocopy inserted prior to Equus divergence. Approximately unbiased p-values indicating the strength of the clustering are shown at each node. *LCORL* parent gene sequences were obtained from three recent equine genome assemblies: EquCab3.0 (*E. caballus*), UCLA_HA_Equagga_1.0 (*E. quagga*) and ASM1607732v2 (*E. asinus*). (**B**) WGS reads spanning an *LCORL* exon-exon junction indicative of the presence of a retrocopy are observed in all Equids (top) while absent from tapirs and rhinoceroses (bottom).

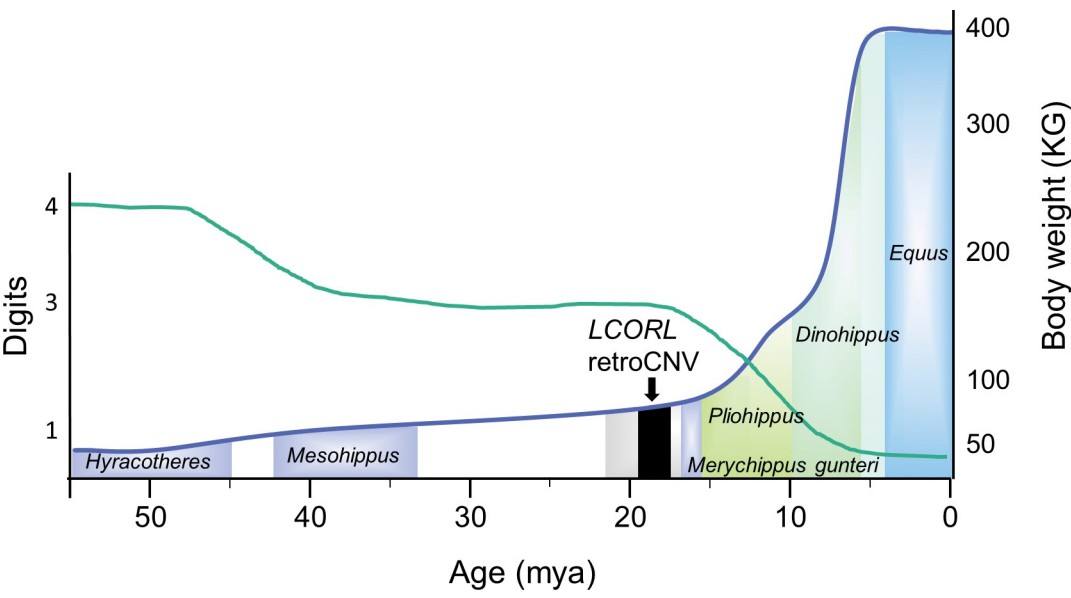

**Fig 5. Insertion of the *LCORL* retrocopy within the timeline of equid evolution.** The body weight (blue line) and digit number (green line) are shown relative to the evolutionary history of equids. Additional changes in tooth anatomy are not shown. There is overlap of different equid species which is shown by transparency of the bars representing the different species. The black bar represents the average age estimate of the *LCORL* retrocopy insertion with 95% confidence interval. The grey bar shows the complete range of insertion times.

ranged from 12.2 to 21.7 MYA, with an average age estimate of 18.0 MYA (95% CI 17.0–19.0 MYA). The age of the retrotransposition placed this event much deeper than the branching of the extant Equus genus, which lead us to explore the timing relative to evolutionary changes that occurred within the Equidae family. The estimate of the time of the *LCORL* retrocopy insertion was compared to the increase in body weight, decrease in digit number, and continued increase in the size of the middle phalanx during equid evolution (Fig 5).

## Discussion

In this study, we applied a recently developed approach to retroCNV discovery to equid WGS data to successfully resolve the insertion sites of 437 retroCNVs. Horses had on average 22.5 retroCNVs, and 219 retroCNVs were specific to horses. While the wild equids had a larger number of retroCNVs on average (61.1), this is likely due to the use of a horse reference assembly for discovery. When retrocopies present in the EquCab3.0 reference assembly itself were also analyzed, 32 retrocopies were found to be absent from all wild equids, identifying them as horse specific retroCNVs. Of particular interest was the identification of a large number of *LCORL* retrocopies in all equids. The *LCORL* retrocopies were also shown to be highly expressed in all tissue types analyzed in both horse and donkey RNAseq data. They are part of a segmental duplication of which copy number was estimated to vary between species and individuals. Based on SNVs between the *LCORL* parent gene and the retrocopies, the retroCNV insertion, common to all extant equids, was dated to 18 MYA. Both the age of the *LCORL* retrogene and the conservation of a large number of expressed *LCORL* retrocopies in all extant equids provides evidence of function for the *LCORL* retrocopies in equid evolution.

A recent study which used the same method to identify retroCNVs in human and dog genomes found on average 4.2 retroCNVs in humans, while dogs had on average 54.1 retroCNVs [17]. In dogs, the increased rate in gene retrotransposition relative to humans was

correlated with increased LINE-1 and SINE dimorphism rates, possibly indicative of higher LINE-1 activity in general [72, 73]. The domestic horse, with 22.5 retroCNVs on average, appears to have an intermediate rate of gene retrotransposition between that of humans and dogs. An analysis of the horse genome identified only 72 full length, intact LINE-1 sequences, which is fewer than both humans, which have 142, and dogs, which have 264 [11], indicating that while horses have fewer total LINE-1, LINE-1 appears to be more actively capable of gene retrotransposition in horses than in humans. As retroCNVs are only one aspect of overall LINE-1 activity, a proportional increase in frequency for LINE-1 and SINE polymorphisms could be expected in equine genomes relative to humans. However, phenotypic associations with transposable element insertions are relatively rare in horses, being responsible for less than 1% (1/103) of the total phenotype associated variants identified in horses, compared to around 10% in dogs (Online Mendelian Inheritance in Animals, OMIA. Sydney School of Veterinary Science, 10/18/2022. World Wide Web URL: https://omia.org/).

RetroCNVs, which are a form of large structural variant, can be difficult to identify from Illumina short read datasets. The retroCNVs identified in low coverage human genomes have varied based on the approach used [2, 10, 13, 74]. Some approaches to retroCNV discovery also focus on retroCNV parental gene analysis [16], but the resolution of the retroCNV insertion site is important for downstream functional analyses. Additionally, multiple insertions can exist for a single parent gene; in this study, 437 total insertions were identified from 353 parent genes. While a previous analysis using multiple long-read human genome assemblies appeared to give the most accurate results for retroCNVs, data of this quality is not yet available for most species [15]. The approach to retroCNV discovery employed in the current study was also used in canids, where it was tested using long-read assemblies and accurately identified 286 out of 292 retroCNVs [17]. A previous study focused on Thoroughbreds found 62 retroCNV parent genes which were PCR validated [18]; the current study confirmed the presence of 54 of those retroCNVs, including the identification of the insertion sites.

The GO enrichment analysis of the retroCNV parent genes showed that many were derived from genes encoding ribosomal proteins, which are known to be common retrocopy forming genes [75]. Retrocopies tend to derive from highly expressed genes, including genes encoding ribosomal proteins [10, 75, 76]. However, the retroCNVs identified in this study may function through a variety of mechanisms, including alteration of nearby gene expression [20, 21]. In natural populations of mice, many retroCNVs were shown to significantly alter the overall expression of the parent gene, and the retroCNVs were also under negative selection, likely due to deleterious effects [14]. Still, many retroCNVs appear to have been selected towards fixation in horses; 21 retrocopies were identified in the EquCab3.0 genome assembly that were present in a homozygous state in all horses yet absent from all wild equids.

More intriguing still is the presence of many *LCORL* retrocopies across all equids. The *LCORL* retrocopies, which are part of a larger duplication that is poorly assembled in most available equid genome assemblies, were only identified due to their absence from the EquCab3.0 assembly. In humans, 7% of the human genome consists of segmental duplications and, in the past, they were also poorly resolved in genome assemblies [77]. The *LCORL* retrocopies were previously mapped to horse chromosome 9 using fluorescence *in situ* hybridization, where it was also noted that very strong signals indicated the presence of multiple copies [51]. While FISH is a good method to visualize the chromosomal location (metaphase analysis) and evaluate approximate copy numbers (interphase analysis), the resolution of FISH is not sufficient for accurate copy number quantitation. This is further confounded by the fact that some interphase nuclei are captured before DNA replication, and others after DNA replication. FISH analysis also indicates that there are no other chromosomes with multiple *LCORL* retrocopies in equid genomes, however there are two distinct *LCORL* retrocopy clusters,

separated by the centromere, on horse chr9 and the corresponding equid homologs (Fig 2, S2, S3 Figs). Discordant read mapping confirmed that the *LCORL* retrocopies were on chromosome 9. Increased coverage over the segmental duplication containing the *LCORL* retrocopies confirmed the presence of many copies, with horses having 25 on average warranting further investigation of this locus and potential functional effects of these retrocopies.

The *LCORL* retrocopies are of particular interest due to the nature of the gene; the *LCORL* locus has been associated with body size across many mammalian species, including humans and horses, although causative variants have been rarely reported [51, 65–67, 69, 71, 78]. While *LCORL* encodes a putative transcription factor, the function of *LCORL* is not well understood [79]. Alternative splicing of the *LCORL* gene results in multiple transcript variants as well as multiple different protein isoforms [69]. *LCORL* has been identified as a component of polycomb repressive complex 2 which alters methylation and gene transcription of polycomb target genes [80]. It is interesting to speculate that *LCORL* mediates changes in body size through alterations in expression of multiple target genes. The equid *LCORL* retrocopies have accumulated numerous variants, including frameshifts and a predicted nonsense variant in exon 4. However, the *LCORL* retrocopies were shown to be highly expressed in horse and donkey tissues, where the retrocopies made up the vast majority of *LCORL* transcripts. Even if they no longer code for a functional protein, the *LCORL* retrocopy transcripts, which can form chimeric transcripts with the nearby *MRPL15* gene, may have an effect on expression or translation of the parent genes [20, 21]. The presence of these highly expressed *LCORL* retrocopies may also have interfered with functional analyses into the *LCORL* parent gene [65, 81]. Additionally, the large copy number of the *MRPL15* gene itself may also have functional consequences in equids. *MRPL15* is a mitochondrial ribosomal protein that plays a role in protein synthesis within the mitochondria which has been identified as a biomarker for cancers [82–84]. Overall expression of *MRPL15* was increased in horse and donkey tissues relative to humans, likely because of the high copy number of the gene. While the association between *LCORL* and body size make the *LCORL* retrocopies intriguing, *MRPL15* could also be contributing to the function of the segmental duplication, or the chimeric reads between the two genes could be driving the selection and conservation of this segmental duplication.

The *LCORL* retrocopies are absent from rhinoceros and tapir genomes, the two other extant families of the Perissodactyla (odd-toed ungulates) order, making the retrocopies unique to the equid family, the only living monodactylous animals [85]. While the *LCORL* retrocopies are present in all equid species, the retrocopy sequences have diverged; separate variants have become fixed across all retrocopies differentially between the species. This is possible evidence of gene conversion due to the effects of concerted evolution, a process which results in greater sequence similarity of repetitive elements within than across species [86]. This, in addition to the across-species preservation of large copy numbers of the *LCORL* retrocopies, implies a functional role of the segmental duplication. Notably, a similar process of retrocopy insertion followed by segmental duplication has resulted in a large number of partial *TP53* gene retrocopies in elephants [22]. Although many of the *TP53* retrocopies are truncated and no longer functional, they have been under positive selection due to a predicted role in cancer resistance among elephants, although this conclusion has been brought into question recently [87]. Another example of retrocopies leading to expanded gene copy number is for the APOBEC3 antiviral proteins in primates [88]. It seems possible that the *LCORL* retrocopies play a similarly important role in the evolutionary history of the equids.

The *LCORL* retrotransposition was estimated to occur 18 MYA based on sequence divergence between the parent genes and the retrocopy sequences available in assemblies from horse, donkey, and zebra. The retrotransposition event happened first, followed by segmental duplication events and additional subsequent rearrangements within the extant equids. In

addition to preservation of high copy number of the segmental duplication, there has been high expression levels maintained across the extant equids during a time frame of almost 5 million years. The large size and complex nature of the segmental duplication make it challenging to completely resolve and as a result the functionality of this highly transcribed locus is difficult to test. Without DNA sequence data from other extinct equids, it is also difficult to verify the timing of the retrotransposition event or the expansion in copy number. The complete landscape of segmental duplications as well as other evolutionary changes during equid evolution when identified may provide additional insight into the importance of this locus. Nonetheless, the timing of the retrotransposition of *LCORL* predating the major morphological changes in equids is compelling.

## Supporting information

**S1 Fig. Large number of expressed *LCORL* retrocopies in equids.** A segmental duplication encompassing the MRPL15 gene on chromosome 9 contains the *LCORL* retrocopy insertion at chr9:30194359–30194380. Colors highlight the presence of SNV and indels within the duplication.
(TIF)

**S2 Fig. FISH for *LCORL* retrocopies in different horse breeds.** FISH with *LCORL* retrocopies containing horse BAC clone 164B3 on metaphase and interphase chromosomes of four different horse breeds shows LCORL copy number variation between individuals. FISH using the horse BAC clone 79K4 containing the single copy gene PLCG2 are shown for contrast.
(TIF)

**S3 Fig. FISH for *LCORL* retrocopies in different equids.** FISH with *LCORL* retrocopies containing horse BAC clone 164B3 on metaphase and interphase chromosomes of two donkeys and two hinnies showing *LCORL* copy number variation between individual donkeys and the donkey and the horse. Note that in the hinnies, the chromosomes with less copies is the donkey chr12 and the chromosome with more copies is the horse chr9.
(TIF)

**S4 Fig. *LCORL* expression analysis in expanded set of horse tissues.** Percentages of total *LCORL* transcripts derived from the retroCNV is shown for each tissue type. The *LCORL* retrocopy comprised the majority of overall *LCORL* transcripts in all tissue types except testis.
(TIF)

**S5 Fig. *LCORL* retrocopies are highly expressed in donkey tissue.** The *LCORL* retrocopy comprised the majority of overall LCORL transcripts in all tissue types except testis.
(TIF)

**S6 Fig. Expression of the *LCORL* parent gene in humans and horses.** Expression of the parent gene in horses is comparable to human *LCORL* across tissue types.
(TIF)

**S7 Fig. Comparison of *MRPL15* expression between humans and horses.** Expression of the segmentally duplicated *MRPL15* gene is increased in horses relative to humans.
(TIF)

**S8 Fig. Expression of *LCORL* and *MRPL15* in horses and donkeys.**
(TIF)

**S9 Fig. Long read RNAseq data viewed at the *LCORL* retrocopy insertion site (EquCab3.0 chr9:30194359–30194380).** The soft-clipped bases at the insertion site indicate sequences

which match the *LCORL* parent gene on chr3. Chimeric reads were observed between the *LCORL* retrocopy and the nearby MRPL15 gene. Antisense reads highlighted in red were also observed at the insertion site.
(TIF)

**S10 Fig. FISH with a horse BAC clone 164B3 in a white rhinoceros.** Hybridization signals were absent in the rhino, indicating the absence of *LCORL* retrocopies.
(TIF)

**S1 Table. Study dataset.**
(XLSX)

**S2 Table. 5' and 3' junction sequence of the equine retroCNV insertions resolved by TEB-reak.**
(XLSX)

**S3 Table. List of primers used for genotyping equine retroCNV.** Each retroCNV has a separate 3 primer PCR reaction for the 5' and 3' junctions.
(XLSX)

**S4 Table. Non-reference retroCNV identified in 86 equids.**
(XLSX)

**S5 Table. Genotyping results from eight retroCNV in a set of Thoroughbreds.**
(XLSX)

**S6 Table. Location and consequence of retrocopy specific variants.**
(XLSX)

**S7 Table. RetroCNV parent genes with unresolved retrocopy insertions.**
(XLSX)

**S8 Table. EquCab3.0 reference retroCNV.**
(XLSX)

**S9 Table. Copy number of the *LCORL* retrocopies based on increased coverage.**
(XLSX)

**S10 Table. BLAST results for the *LCORL* parent gene sequence in alternative equine genome assemblies.**
(XLSX)

**S11 Table. Variants within the *LCORL* retrocopies.**
(XLSX)

**S12 Table. *LCORL* retrocopy age estimate.** *LCORL* retrocopy sequences within 3 equine assemblies were compared to their parent gene sequence in order to estimate the age of the retrocopy insertion.
(XLSX)

**S1 File. Predicted amino acid sequence of the *LCORL* parent gene and retrocopies.**
(TXT)

**S2 File. *LCORL* sequence with the horse retrocopy specific variants masked.**
(TXT)

**S3 File.** *LCORL* **sequence with the donkey retrocopy specific variants masked.**
(TXT)

**S4 File. Spliced sequence of exon 5 and exon 6 of the** *LCORL* **gene to be used in BLAST.**
(TXT)

## Author Contributions

**Conceptualization:** Peter Dickinson, Danika Bannasch.

**Data curation:** Kevin Batcher.

**Formal analysis:** Kevin Batcher.

**Funding acquisition:** Danika Bannasch.

**Investigation:** Kevin Batcher, Scarlett Varney, Terje Raudsepp, Matthew Jevit, Vidhya Jagan-nathan, Tosso Leeb.

**Methodology:** Kevin Batcher, Danika Bannasch.

**Project administration:** Danika Bannasch.

**Resources:** Danika Bannasch.

**Supervision:** Danika Bannasch.

**Visualization:** Kevin Batcher, Terje Raudsepp, Matthew Jevit.

**Writing – original draft:** Kevin Batcher.

**Writing – review & editing:** Terje Raudsepp, Peter Dickinson, Tosso Leeb, Danika Bannasch.

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
