## [Decision Letter · Decision Letter 0]

11 May 2023

PONE-D-23-11627Ancient segmentally duplicated LCORL retrocopies in equidsPLOS ONE

Dear Dr. Bannasch,

Thank you for submitting your manuscript to PLOS ONE. After careful consideration, we feel that it has merit but does not fully meet PLOS ONE’s publication criteria as it currently stands. Therefore, we invite you to submit a revised version of the manuscript that addresses the points raised during the review process. Please submit your revised manuscript by Jun 25 2023 11:59PM. If you will need more time than this to complete your revisions, please reply to this message or contact the journal office at plosone@plos.org. Please include the following items when submitting your revised manuscript:A rebuttal letter that responds to each point raised by the academic editor and reviewer(s). You should upload this letter as a separate file labeled 'Response to Reviewers'.A marked-up copy of your manuscript that highlights changes made to the original version. You should upload this as a separate file labeled 'Revised Manuscript with Track Changes'.An unmarked version of your revised paper without tracked changes. You should upload this as a separate file labeled 'Manuscript'.If applicable, we recommend that you deposit your laboratory protocols in protocols.io to enhance the reproducibility of your results. Protocols.io assigns your protocol its own identifier (DOI) so that it can be cited independently in the future. For instructions see: https://journals.plos.org/plosone/s/submission-guidelines#loc-laboratory-protocols. Additionally, PLOS ONE offers an option for publishing peer-reviewed Lab Protocol articles, which describe protocols hosted on protocols.io. Read more information on sharing protocols at https://plos.org/protocols?utm_medium=editorial-email&utm_source=authorletters&utm_campaign=protocols.

We look forward to receiving your revised manuscript.

Kind regards,

Ruslan Kalendar

Academic Editor

PLOS ONE

Journal Requirements:

   "This work was funded in part by the Maxine Adler Endowed Chair Funds. "

3. Please note that funding information should not appear in any section or other areas of your manuscript. We will only publish funding information present in the Funding Statement section of the online submission form. Please remove any funding-related text from the manuscript. "Maxine Adler Endowed Chair Funds"

Reviewers' comments:

Reviewer's Responses to Questions

**Comments to the Author**

1. Is the manuscript technically sound, and do the data support the conclusions?

Reviewer #1: Yes

Reviewer #2: Yes

2. Has the statistical analysis been performed appropriately and rigorously? 

Reviewer #1: Yes

Reviewer #2: Yes

3. Have the authors made all data underlying the findings in their manuscript fully available?

Reviewer #1: Yes

Reviewer #2: Yes

4. Is the manuscript presented in an intelligible fashion and written in standard English?

Reviewer #1: Yes

Reviewer #2: Yes

5. Review Comments to the Author

Reviewer #1: 

In this study, Batcher et al. investigated the retro copies and retro copy number variants in domesticated horses and other species in the family Equidae and identified several retrocopies of the gene LCORL that seems to have gone through additional segmental duplication. This study should be of interest to the readers of PLOS One. The manuscript is well written, and the results are intriguing. I highlighted my suggestions for improving the manuscript below.

1. One issue that may confuse the readers is the author’s usage of the terms retrocopies and retroCNVs. Based on the authors’ description in Line 44-45, retroCNVs refer to the retrocopies of genes that are different in number between different individuals of the same species. However, in some parts of the manuscript authors refer to retrocopies that are shared between different species of equids as retroCNVs. An example of this is in the abstract “Only 5 retroCNVs were shared between horses and other equids”. Moreover in the case of the wild species of the Equus genus, in most cases authors only have access to the sequence of a single individual. Based on the definition provided for retroCNV in the introduction, retroCNV identification would not be possible when looking at only a single individual of a species.

2. For clarification purposes for the readers, in the first mention of the word Equid in the main text at line 79, authors should indicate that this refers to the family Equidae.

3. At the end of the sentence at line 80, there should be citation

4. Line 97. Authors refer to the reference assembly. I’m assuming this is the reference assembly for E. caballus. Authors should indicate this, since 3 other species in this genus have reference assemblies.

5. Line 237 indicates that there are 213 retroCNVs identified in domesticated horses but 219 is listed in Table 1.

6. Authors indicate in the abstract and line 276 of results that 5 retroCNVs are shared between wild equids and domesticated horses but only talk about the LCORL retrocopies. I couldn’t find the other shared retrocopies. Readers may be confused after seeing this result in the abstract when it is not clearly presented in the manuscript.

7. Line 327, authors refer to the transcript variant of Lcorl in mice as “observed”. However NCBI XM_ designation in transcripts indicate computer annotated transcripts, not experimentally validated ones.

8. Tables S6-12 are mislabeled in the excel file and Table S11 is missing the title.

9. Figure S9. Menu items, zoom levels and file name should be removed

10. Figure 4A. Authors should indicate what kind of sequence they used to generate this phylogenetic tree (DNA or amino acid) in the figure legend. I also recommend providing the sequences used to generate this tree as a supplementary file.

11. Authors should indicate what different colors mean in the alignments they show in Figure 2 Top, Figure 3A and Figure S1. Only the meaning of red color is indicated in Figure 2 legend.

12. Authors should indicate the actual source of the thoroughbred horse samples used in genotyping in the methods section instead of only referring to the DNA repository in their lab.

Reviewer #2: 

Retrocopies of genes, also called processed pseudogenes, are produced by the action of the L1-encoded reverse transcriptase. Some of them are known to be insertionally polymorphic, generating retrocopy CNVs (retroCNVs) in mammalian genomes. The authors recently studied such retroCNVs in canids, and in this manuscript, they identified ~500 retroCNVs in equids (438 not present in the reference genome, and 32 present in the reference) and their genomic insertion sites. The authors then revealed that retrocopies of LCORL are present in a region of segmental duplication and are transcribed ubiquitously. Based on the sequence divergence, the authors estimated the timing of the LCORL retrocopy insertion to be ~18 MYA, before the timing of digit number reduction and body weight increase in equids. This may suggest the involvement of the LCORL retrocopy in such morphological changes. Although the evidence for it is weak, this seems in line with the previous finding that FGF4 retroCNVs are involved in the leg shortening in domesticated dogs (Science 2009, 325:995-998). The computational analyses as well as FISH experiments were appropriately conducted, and the manuscript is well organized. I recommend its publication, while I have several minor comments below.

Minor Comments.

[1] Page2, line 29, “… is the only autonomous transposable element ...” should be "... is one of autonomous transposable element...". (Some non-L1 TEs are active in rodents, cows, bats, etc.)

[2] Page2, line 35, in addition to ref. (4), the following paper can also be cited, which shows that recognizable long TSD is typical for L1 in human, cow, opossum, and zebrafish: Ichiyanagi and Okada (2008) Mol Biol Evol 25, 1148-1157.

[3] In Fig. 3B, “LCORL retroCNV” on the top-right corner should be “LCORL retrocopies”.

[4] The caption of Table S11 is missing, and the caption of Table S12 is wrong.

6. PLOS authors have the option to publish the peer review history of their article (what does this mean?). If published, this will include your full peer review and any attached files.

Reviewer #1: No

Reviewer #2: **Yes: **Kenji Ichiyanagi

---

## [Author Response · Author response to Decision Letter 0]

24 May 2023

Below in italics are our responses to concerns from the academic editor and reviewers: 

Journal Requirements:

The file naming scheme has been corrected to match PLOS ONE’s style requirements throughout the manuscript.

 "This work was funded in part by the Maxine Adler Endowed Chair Funds. "

The funders had no role in the study, and the cover letter has had this statement included.

3. Please note that funding information should not appear in any section or other areas of your manuscript. We will only publish funding information present in the Funding Statement section of the online submission form. Please remove any funding-related text from the manuscript. "Maxine Adler Endowed Chair Funds"

The funding section has been removed from the manuscript.

An Ethics statement has been included in the methods section.

Reviewer #1: 

1. One issue that may confuse the readers is the author’s usage of the terms retrocopies and retroCNVs. Based on the authors’ description in Line 44-45, retroCNVs refer to the retrocopies of genes that are different in number between different individuals of the same species. However, in some parts of the manuscript authors refer to retrocopies that are shared between different species of equids as retroCNVs. An example of this is in the abstract “Only 5 retroCNVs were shared between horses and other equids”. Moreover in the case of the wild species of the Equus genus, in most cases authors only have access to the sequence of a single individual. Based on the definition provided for retroCNV in the introduction, retroCNV identification would not be possible when looking at only a single individual of a species.

Throughout the manuscript, the term retroCNV is used to refer to retrocopies that are polymorphic between any members of the equid family. We have clarified in the introduction that retroCNV can be used to refer to retrocopies that differ both within and between species. 

2. For clarification purposes for the readers, in the first mention of the word Equid in the main text at line 79, authors should indicate that this refers to the family Equidae.

Added for clarification (line 79)

3. At the end of the sentence at line 80, there should be citation

Citation added

4. Line 97. Authors refer to the reference assembly. I’m assuming this is the reference assembly for E. caballus. Authors should indicate this, since 3 other species in this genus have reference assemblies.

Yes this was in reference to the equcab3.0 reference, this been included in the manuscript.

5. Line 237 indicates that there are 213 retroCNVs identified in domesticated horses but 219 is listed in Table 1.

Thank you this was corrected.

6. Authors indicate in the abstract and line 276 of results that 5 retroCNVs are shared between wild equids and domesticated horses but only talk about the LCORL retrocopies. I couldn’t find the other shared retrocopies. Readers may be confused after seeing this result in the abstract when it is not clearly presented in the manuscript.

The other shared RetroCNV are now included in the transcript.

7. Line 327, authors refer to the transcript variant of Lcorl in mice as “observed”. However NCBI XM_ designation in transcripts indicate computer annotated transcripts, not experimentally validated ones.

Thank you this has been corrected. (line 332-333).

8. Tables S6-12 are mislabeled in the excel file and Table S11 is missing the title.

The supplementals have all been updated and corrected throughout the manuscript and the supplemental files. 

9. Figure S9. Menu items, zoom levels and file name should be removed

Thank you, this figure has been updated.

10. Figure 4A. Authors should indicate what kind of sequence they used to generate this phylogenetic tree (DNA or amino acid) in the figure legend. I also recommend providing the sequences used to generate this tree as a supplementary file.

Fig4 figure legend has been changed to indicate this analysis was using DNA; the LCORL SNV used in this analysis are available in supplemental table S6.

11. Authors should indicate what different colors mean in the alignments they show in Figure 2 Top, Figure 3A and Figure S1. Only the meaning of red color is indicated in Figure 2 legend.

The figure legends have been updated to include this information.

12. Authors should indicate the actual source of the thoroughbred horse samples used in genotyping in the methods section instead of only referring to the DNA repository in their lab.

The origin of the DNA samples has been included in the Ethics statement. 

Reviewer #2: 

Minor Comments.

[1] Page2, line 29, “… is the only autonomous transposable element ...” should be "... is one of autonomous transposable element...". (Some non-L1 TEs are active in rodents, cows, bats, etc.)

This has been corrected (line 29).

[2] Page2, line 35, in addition to ref. (4), the following paper can also be cited, which shows that recognizable long TSD is typical for L1 in human, cow, opossum, and zebrafish: Ichiyanagi and Okada (2008) Mol Biol Evol 25, 1148-1157.

Thank you we will include this reference.

[3] In Fig. 3B, “LCORL retroCNV” on the top-right corner should be “LCORL retrocopies”

Corrected.

[4] The caption of Table S11 is missing, and the caption of Table S12 is wrong.

The supplementals have all been updated and corrected throughout the manuscript and the supplemental files.

---

## [Editor Report · Decision Letter 1]

25 May 2023

Ancient segmentally duplicated LCORL retrocopies in equids

PONE-D-23-11627R1

Dear Dr. Bannasch,

We’re pleased to inform you that your manuscript has been judged scientifically suitable for publication and will be formally accepted for publication once it meets all outstanding technical requirements.

Kind regards,

Ruslan Kalendar

Academic Editor

PLOS ONE

---

## [Editor Report · Acceptance letter]

30 May 2023

PONE-D-23-11627R1 

Ancient segmentally duplicated *LCORL* retrocopies in equids 

Dear Dr. Bannasch:

I'm pleased to inform you that your manuscript has been deemed suitable for publication in PLOS ONE. Congratulations! Your manuscript is now with our production department. 

Kind regards, 

on behalf of

Professor Ruslan Kalendar 

Academic Editor

PLOS ONE